# pCTX-M3—Structure, Function, and Evolution of a Multi-Resistance Conjugative Plasmid of a Broad Recipient Range

**DOI:** 10.3390/ijms22094606

**Published:** 2021-04-27

**Authors:** Izabela Kern-Zdanowicz

**Affiliations:** Institute of Biochemistry and Biophysics, Polish Academy of Sciences, 02-106 Warsaw, Poland; iza@ibb.waw.pl; Tel.: +48-22-592-1206

**Keywords:** plasmid, IncM, *bla*_CTX-M-3_, conjugative transfer, antibiotic resistance

## Abstract

pCTX-M3 is the archetypic member of the IncM incompatibility group of conjugative plasmids (recently referred to as IncM2). It is responsible for the worldwide dissemination of numerous antibiotic resistance genes, including those coding for extended-spectrum β-lactamases and conferring resistance to aminoglycosides. The IncM plasmids acquired during evolution diverse mobile genetic elements found in one or two multiple resistance regions, MRR(s), grouping antibiotic resistance genes as well as mobile genetic elements or their remnants. The IncM plasmids can be found in bacteria inhabiting various environments. The information on the structure and biology of pCTX-M3 is integrated in this review. It focuses on the functional modules of pCTX-M3 responsible for its replication, stable maintenance, and conjugative transfer, indicating that the host range of the pCTX-M3 replicon is limited to representatives of the family Enterobacteriaceae (Enterobacterales ord. nov.), while the range of recipients of its conjugation system is wide, comprising Alpha-, Beta-, and Gammaproteobacteria, and also Firmicutes.

## 1. Introduction

Plasmids are DNA molecules able to replicate independently from the host-cell chromosome, widespread in bacteria and archaea, but also found in some eukaryotes. They have a modular structure comprising a backbone and a load. The backbone is composed of a replicon, regions encoding maintenance functions (oligomer resolution, partition, and post-segregational killing), and conjugative transfer genes. The plasmid load comprises all DNA fragments gathered during plasmid evolution, encoding, for example, antibiotic resistance, metabolic pathways, or virulence factors. 

Plasmids are part of a mobile gene pool available for bacteria. Especially in Gram-negative bacteria, plasmids transferred during conjugation play an important role, being responsible for the bacterial ability to adapt quickly to a changing environment. Of particular importance are plasmids involved in the spreading of antimicrobial resistance in clinically relevant bacterial species. Several bacteria resistant to last resort antibiotics were listed in 2017 by the World Health Organization as those of critical priority for research and development of new antibiotics (https://www.who.int, accessed on 20 November 2020). Beside carbapenem-resistant *Acinetobacter baumannii*, *Pseudomonas aeruginosa*, and species of the Enterobacteriaceae family (recently reclassified as order Enterobacterales [1]), this priority also concerns Enterobacteriaceae species producing extended-spectrum β-lactamases (ESBL). According to the definition by D. Livermore, “an ESBL is any β-lactamase, ordinarily acquired and not inherent to a species, that can rapidly hydrolyze, or confer resistance to, oxyimino-cephalosporins (not carbapenems) or any β-lactamase mutant, within a family, that has an enhanced ability to do so” [2].

The ESBLs encountered in Europe are variants of β-lactamase types SHV, TEM, and CTX-M, with the latter widely disseminated in the 2000s [3]. The first such enzyme was detected in Munich, Germany in 1989 as conferring resistance to cefotaxime in *Escherichia coli* and therefore was named CTX-M for cefotaximase-Munich [4]. Since then, more than 300 CTX-M β-lactamases have been described, classified into seven groups based on their phylogeny: CTX-M-1, -M-2, -M-8, -M-9, -M-14, -M-15, and -M-25. Genes encoding enzymes of the first two groups evolved by escape of chromosomal genes from *Kluyvera ascorbata*, and those of groups 8 and 9 from *K. georgiana* [5,6]. The insertion sequence IS*Ecp1* was often involved in the initial mobilization of *bla*_CTX-M_ ancestors from the genome of *Kluyvera* sp., and is frequently found in the proximity of the *bla*_CTX-M_ genes on plasmids. Moreover, the IS*Ecp1* element can also be responsible for the high-level expression of the *bla*_CTX-M_ genes [5].

The mobilization of the CTX-M-encoding genes into large conjugative plasmids has brought about their global dissemination. Around the year 2000, the CTX-M-9 and CTX-M-14 variants were the most prevalent in Spain, CTX-M-1—in Italy, CTX-M-3 – in Poland, and the CTX-M-15 variant in France and the UK ([3] and citations therein). Plasmids responsible for the spread of the *bla*_CTX-M_ gene variants often belong to one of six major resistance plasmid families identified in clinically relevant Enterobacteriaceae (for review see [7,8]). Worldwide, the most frequently isolated epidemic clone was *E. coli* ST131 with *bla*_CTX-M-15_ located on a plasmid of the IncF incompatibility group [9]. 

## 2. Isolation of pCTX-M3

The “Polish CTX-M variant” was detected in 1996 in a hospital in Warsaw, when three *Citrobacter freundii* strains and one *Escherichia coli* strain resistant to cefotaxime and susceptible to ceftazidime were isolated from urine samples of different patients [10]. The resistance was associated with a large plasmid coding for a CTX-M variant new at that time, CTX-M-3. This plasmid could be easily transferred to *E. coli* via conjugation and conferred resistance to penicillins, cephalosporins, and aztreonam, as well as to the aminoglycosides, gentamicin and tobramycin, and trimethoprim-sulfamethoxazole [10]. Between 1998 and 2000 bacteria carrying this plasmid became widely disseminated in Poland and they were found in 15 hospitals in ten Polish cities; they represented eight Enterobacteriaceae species: *Klebsiella pneumoniae*, *K. oxytoca*, *Enterobacter cloacae, Serratia marcescens*, *C. freundii, E. coli, Salmonella enterica*, and *Morganella morganii* [11]. The nucleotide sequence of the plasmid isolated from the clinical strain *C. freundii* 2526, and subsequently named pCTX-M3, has been determined and deposited in the GenBank database under the accession no. AF550415.

## 3. Analysis of pCTX-M3 Nucleotide Sequence

The pCTX-M3 plasmid is 89,468 bps in size and carries 103 putative genes, of which 22 *orfs* have no homologues in public databases and are regarded as pCTX-M3-specific [12]. 

The mean G+C content in pCTX-M3 DNA is 51%, but it varies from 32% up to 65% in different sequence blocks (Figure 1). The minimal G+C value is found in two regions, one comprising *armA*, *mel*, *mph1*, and *mph2*, and the other within IS*Ecp1* and *bla*_CTX-M-3_, while the maximum in the sequence around *orf8*, a pCTX-M3-specific gene. Antibiotic resistance genes are located in two horizontally acquired regions, the first comprising *bla*_CTX-M-3_ with IS*Ecp1*, responsible for its introduction to the plasmid, and the second, a multiple resistance region (MRR). It comprises *bla*_TEM-1_, situated next to the replicon, a type I integron with the *aadA2*, *dfrA12*, and *sul1* gene cassettes, and the low-(G+C) region mentioned above, with *armA* coding for a 16S rRNA methylase and three putative macrolide resistance genes *mel*, *mph1*, and *mph2*. The MRR lies between the replicon and the *trb* transfer region; it is 27 kb in size and contains all the mobile elements of pCTX-M3 except for IS*Ecp1* [12]. Moreover, MRR has a mosaic structure with areas of high and low G+C contents (Figure 1), indicating that it arose by multiple genetic events, of which the first was acquisition of Tn*1* together with *bla*_TEM-1_ [12]. It is worth mentioning that *armA*, *mel*, *mph1*, *mph2*, and the integron are located within Tn*1548*, a composite transposon bordered by two IS*26* copies [13], also comprising the IS*CR1* element regarded as a gene-capturing system (for review see [14]). It has been speculated that Tn*1548* gave rise to the AbGRI3 resistance islands of *Acinetobacter baumannii* [15].

### 3.1. Replicon

pCTX-M3 replicates owing to a single replicon in 96% identical to that of pMU604 (mini-pMU607.1) of *K. pneumoniae* (GenBank acc. no. U27345) and comprise the *repCBA* region. *repA* codes for a replication initiator protein and *repB* for a small leader peptide translationally coupled with RepA [16], while the function of *repC* is unclear. The regulation of expression of the replication region in the pMU604 plasmid was deciphered by Athanasopoulos et al. [16] and found to be mainly post-transcriptional. The expression of *repB* and *repA* is repressed by an antisense RNA of ca. 77 nucleotides, called RNA I, whose target lies in *repB*, the leader region of the *rep* mRNA. Expression of *repA* is dependent on *rep* mRNA forming a tertiary structure called a pseudoknot, which enables translation of *repA* mRNA. Interaction of RNA I with its target sequesters the mRNA bases involved in the formation of the pseudoknot, and thereby prevents *repA* expression [16,17]. 

In *E. coli* cells pCTX-M3 is present in approx. 0.5–0.7 copies per chromosome (measured in cells of early stationary culture), while a minireplicon, comprising the *repCBA* region, is present in 8–10 copies per chromosome [18]. 

The pCTX-M3 replicon was originally classified as a member of an IncL/M incompatibility group [12]. However, in 2015 [19] three IncL/M plasmid lines, IncL, IncM1, and IncM2, were distinguished on the basis of their *inc*RNA identity, which is 104 bases long [20] and partially overlaps the RNA I described by Athanasopoulos et al. [16]. Representatives of the IncL and IncM plasmids have been shown to be mutually compatible, unlike members of the IncM1 and IncM2 groups. pCTX-M3 was reclassified as an IncM2 group plasmid [19].

The range of hosts in which pCTX-M3 can replicate comprises Enterobacteriaceae (Enterobacterales ord. nov.) [21].

### 3.2. Stable Maintenance Systems

pCTX-M3 is stably maintained in bacterial populations (it is carried by 100% *E. coli* cells after 60 generations [18]). A stable maintenance of a plasmid in a bacterial population is ensured by a concerted action of the partition system, post-segregational killing system, resolvase, and finally by the conjugation system. The maintenance of the pCTX-M3 plasmid is ensured by the *parA–parB* module coding for a partition system homologous to the *parM–parR*-encoded system from the R1/NR1 plasmid [12], with the first gene coding for an actin-type segregation protein and the second for a DNA-binding adaptor protein (for review see [22]). In pCTX-M3 the centromere-like sequence *parS* is located upstream of *parA*, similarly to *parC*, the centromere in the *parMR* system [18]. Another component has been shown to be important in the pCTX-M3 partition system, playing a role in the stabilization of unstable vectors—the *nuc* gene located just downstream of *parB* and coding for a nuclease [18]. In IncI1 plasmids, such as R64, *nuc* is located within the *tra* region, upstream of *sogL* coding for primase. However, its inactivation had no effect on the conjugative transfer of this plasmid [23]. Resolvase, an enzyme resolving plasmid oligomers into monomeric forms, is probably encoded by the *resD* gene in pCTX-M3 [12]. A post-segregational killing (PSK) system is encoded by the *pemI–pemK* operon showing high similarity to that of NR1 (97% identity of PemI and 95% identity of PemK) [12]. *pemK* codes for a toxin, which was shown to be a sequence-specific endoribonuclease cleaving mRNAs to inhibit protein synthesis, and *pemI* for its antidote [24]. Deletion of *pemI–pemK* resulted in only a small drop in pCTX-M3 maintenance in *E. coli* cells (97% retention of the PSK system-deficient plasmid after 60 generations), indicating the unimportance of the PSK system for the segregational stability of pCTX-M3 and the main role of the partition system [18].

The pCTX-M3 conjugation system is encoded in two distantly located regions, *tra* and *trb*, both displaying an operon structure. It is related to that of IncI1 plasmids, ColIb-P9 and R64, in both the amino acid sequences of respective proteins (30 to 65% similarity of amino acid sequence) and in the gene syntheny (Figure 2) [12]. Nevertheless, several genes from each system are unique. The two systems also differ in the location of entire gene segments—in pCTX-M3, the *oriT* region (the origin of transfer sequence *oriT* together with *nikA* and *nikB*, coding for an accessory protein and a relaxase, respectively) is a part of the *tra* operon, while in the IncI1 plasmids it is located downstream of the *trb* cluster and the two regions are transcribed convergently. Moreover, there is no *traEFG* in pCTX-M3, and its *tra* and *trb* regions are separated by the IncM replicon and the 27-kb MRR. 

The *oriT*_pCTX-M3_ sequence lies upstream of *nikA* and comprises two pairs of inverted repeats and the *nic* site ACATCTTG↓T similar to that of the IncI1 plasmids [25]. Moreover, pCTX-M3 can mobilize plasmids bearing *oriT* of the IncI1 plasmid ColIb-P9, and ColIb-P9 can mobilize plasmids bearing *oriT*_pCTX-M3_, albeit both with low efficiency [12]. 

pCTX-M3 can be transferred during bacterial conjugation not only on solid supports, but also in liquid. Interestingly, in contrast to the IncI1 plasmids which require type IV pili for conjugation in liquid media, pCTX-M3 does not encode additional pili [12,26], similarly to other IncM and IncL plasmids [27].

#### 3.2.1. pCTX-M3 Conjugative Transfer System

The conjugative transfer can be regarded as a rolling circle (RC) replication coupled with a type IV protein transport system (T4SS) by a dedicated coupling protein (CP) [28]. During conjugation of bacteria other than *Streptomyces*, a plasmid or an integrative conjugative element (ICE) is transported as single-stranded DNA (ssDNA) complexed with specific proteins. The DNA transfer occurs after a physical contact between the donor and the recipient cell is established thanks to the activity of the MPF (*mating pair formation*) system, a secretion machinery for DNA–protein complexes, homologous with the type IV secretion system (T4SS). During bacterial conjugation the plasmid DNA is processed by the relaxosome complex (also called DTR—DNA transfer and replication). Its action involves the nicking of one strand of the plasmid DNA at a specific sequence, the *nic* site, located within the origin of transfer (*oriT*). The free 3’OH end of the nicked strand then serves as a primer for the synthesis of a new DNA strand in a process resembling RC replication. The relaxosome complex is composed of the relaxase, which is an ssDNA transesterase, and auxiliary proteins [29]. The relaxase remains covalently bound to the 5’ end of the ssDNA and is transported to the recipient cell. The DNA segment entering the recipient cell the earliest is termed the plasmid leading region. The coupling protein (CP) bringing together the action of the DTR and MPF systems is an ATPase, and it is also involved in targeting the transported plasmid DNA in the form of a nucleoprotein to the MPF system [30]. In the recipient cell, the ssDNA is circularized and then copied to form dsDNA [29]. 

The MPF systems of conjugative plasmids are evolutionarily related to the T4SS which transports virulence proteins of some pathogenic Gram-negative bacteria into eukaryotic cells [31]. The prototypical T4SS is the VirB/VirD4 transporter transferring the oncogenic T-DNA from *Agrobacterium tumefaciens* into plant cells. This T4SS consists of 11 proteins (VirB1 to VirB11), and VirD4 being the CP (for review, see [32,33]). The translocation channel comprises the VirB3 and VirB6–VirB10 proteins. Three components form the core channel complex in the outer membrane, also called the outer membrane complex (OMC): VirB9, the pore-forming protein; VirB7, a small lipoprotein; and VirB10, spanning both the inner and the outer membrane. The inner membrane complex (IMC) comprises the VirB3, VirB6, and VirB8 proteins. Interactions of the IMC with OMC and with the ATPases VirB4 and VirB11 result in the formation of a pore. The extracellular structure important for establishing the contact between the donor and recipient cells, the T-pilus, consists of VirB2, the major subunit, and VirB5, the minor component localized at the pilus tip. Additionally required for the T-pilus assembly is VirB3, the least-characterized MPF component. Finally, VirB1 shows homology to a lytic transglycosylase that cleaves peptidoglycan [34,35]. Three cytoplasmic ATPases, VirB4, VirB11, and the coupling protein VirD4, provide energy for the system.

The T4SSs of Gram-negative bacteria fall into two large phylogenetic groups, IVA and IVB [36]. The VirB/VirD4 transporter of *A. tumefaciens* is the prototype of type IVA systems (T4ASS) [32]. The type IVB group (T4BSS) comprises homologues of the Dot/Icm transporter of virulence proteins of *Legionella pneumophila*, the causative agent of Legionnaires’ disease, composed of approximately 27 proteins [34,37]. Most of the Dot/Icm proteins share homology with the elements of conjugation system of the IncI1 plasmid R64 [38] (Figure 2). 

The homology between the proteins involved in R64 conjugative transfer and the components of the VirB/VirD4 transporter of *A. tumefaciens* is rather low [39]. The R64 conjugative transfer system is encoded by 22 genes from the *tra* region, *traE–traY* and the *nuc* gene, plus the *trbA–C* genes. Altogether, 16 of them have been shown to be indispensable for conjugation [23]. The conjugative system of pCTX-M3 also comprises *tra* and *trb* genes; however, as already mentioned, the transfer regions of these two plasmids do differ. Nevertheless, the pCTX-M3 proteins, similarly to that of R64, display some homology to a number of Dot/Icm proteins (Figure 2). Results of our bioinformatic analysis (for details see [21]) suggest that the T4SS of the pCTX-M3 conjugation system is composed of:

The transmembrane core subcomplex. Its OMC comprises TraN, TraI, and TraH, the homologues of *Legionella* DotH, DotC, and DotD, respectively. These *Legionella* proteins localize to the outer membrane forming the pore, similar to that formed by the VirB7–VirB9 proteins; by combining with the inner membrane proteins DotF and DotG they form the transmembrane core subcomplex [40]. In pCTX-M3, the DotF and DotG homologues are TraP and TraO; the latter is also a distant homologue of VirB10 [21,41];The pilus—the TraR and TraQ proteins are distant homologues of VirB2, the major pilus subunit;The VirB11 and VirB4 ATPases—TraJ is a homologue of VirB11 and TraU is a distant homologue of *Legionella* DotO and *Agrobacterium* VirB4, the only component common to all T4SSs of Gram-negative and Gram-positive bacteria, and archaea [39]; The coupling protein—the putative CP is TrbC. The CP subcomplex may also comprise TrbA, a DotM homologue;The entry exclusion system—TraY and ExcA of pCTX-M3 are close homologues of respective R64 proteins [42].

A deletion analysis of individual genes from the *tra* and *trb* regions of pCTX-M3 revealed that all but three of them are important for conjugative transfer both in liquid media and on solid surface [21]. It is worth mentioning that deletion of *traI* or *traO* in IncI1 plasmids only diminished the conjugative transfer on solid support, and deletion of *traH* had no influence [23]. In contrast, the deletion of homologues of any of these three genes in pCTX-M3 abolished the transfer, suggesting different compositions of the core transmembrane subcomplexes encoded by these two plasmids. 

The three genes found to be dispensable for the pCTX-M3 conjugative transfer are: *orf36*, located in the middle of the *tra* region, *orf46* from the *trb* region, and *orf35* from the leading region. Interestingly, a pCTX-M3 derivative without *orf35* or *orf36* used as a helper plasmid demonstrated an increased mobilization efficiency [21]. Deletion of *orf35* resulted in a 100-fold increase in mobilization efficiency while the effect of *orf36* deletion was dependent on the recipient species: with *A. tumefaciens* and *E. coli* as recipients, the increase was ca. 10- and 5-fold, respectively. An analysis of transcript levels of the 5′-most genes of the *tra* region of pCTX-M3 lacking *orf35* led to a conclusion that the pCTX-M3 *tra* operon, which encodes both the relaxase complex and T4SS, is subject to *orf35*-dependent repression: expression of *nikA*, *nikB*, and *traH* was elevated ca. 40-, 23-, and 80-fold [21]. Deletion of *orf36* did not change the level of transcripts of *nikA* or *nikB*, while that of *traH* was increased 120-fold, suggesting that the T4SS-encoding genes are controlled by an additional promoter subject to *orf36-*dependent repression. However, the overproduction of the Orf36 protein was not succeeded [43]. It should be noted here that within *orf36* but on the opposite strand two other *orfs* were detected, one coding for an H-NS-like protein and the other an Hha-like protein, both members of the nucleoid-associated proteins (NAP) family. Plasmid-encoded NAPs participate in transcriptional silencing of horizontally acquired genes to prevent a fitness decrease in the host cell [44]. These two NAP-encoding *orfs* can also be identified in other plasmids of the IncL and IncM groups, especially when automatically annotated. They were first described in the R446 plasmid by Tietze and Tschäpe in 1994 [45] who were looking for an *E. coli* mutant that was insensitive to an M-specific phage. Taking into account that *orf36* and NAP-encoding genes are located in the center of the *tra* region, the mode of regulation of the conjugative transfer genes in pCTX-M3 requires further studies.

Not only conjugative plasmids can be transferred during conjugation. Plasmids carrying an *oriT* sequence at least compatible with the MPF encoded by a plasmid co-residing in the host cell can be mobilized to conjugative transfer. The plasmid supplying the MPF is then regarded as a helper plasmid, and the plasmid being transferred with its help is called a mobilizable plasmid.

pCTX-M3 acting as a helper plasmid can mobilize the *oriT*_pCTX-M3_-bearing plasmids into recipients such as *A. tumefaciens*, *Cupriavidus necator* (previously *Ralstonia eutropha*), and *Pseudomonas putida*, representatives of Alpha-, Beta-, and Gammaproteobacteria, respectively [21], and also into the Gram-positive bacteria *Bacillus subtilis* and *Lactococcus lactis* [25]. pCTX-M3 itself, when transferred, cannot be established in any these bacteria [21,25]. These indicate that the range of hosts of the conjugative transfer system of pCTX-M3 is much broader than the host range of its replicon, which is restricted to Enterobacteriaceae (Enterobacterales ord. nov.).

## 4. Evolution of pCTX-M3-Related IncM Plasmids

At the end of the 20th century, 47% of strains of the plant pathogen *Erwinia amylovora*, the causative agent of fire blight, isolated in Lebanon carried a ca. 60-kb plasmid of the IncL/M group, pEL60 [46]. Its backbone is almost identical to that of pCTX-M3, e.g., the identity of the *repCBA* genes is 93%. The main difference between these plasmids is the 29-kb sequence present in pCTX-M3, organized in two regions with antibiotic resistance genes: first, downstream of *pemK*, with IS*Ecp1* introducing *bla*_CTX-M-3_, and the second, the MRR, separating *repA* and the *trb* region. Although pEL60 appeared to be the pCTX-M3 ancestor, it was classified as an IncM1 plasmid [19]. Surprisingly, the nucleotide sequence of a plasmid isolated from *Enterobacter hormaechei* in an Australian hospital, in 2019, pCP15_002 (GenBank acc. no. CP042490), is over 96% identical to that of pEL60. The plasmid carries no antibiotic resistance genes, but its entire sequence can be found in pCTX-M3: it is 98–99% identical to the pCTX-M3 backbone, with the *rep* region identical in 100%. Therefore, pC15_002 could be considered the actual pCTX-M3 ancestor, still circulating in the environment. 

Interestingly, another plasmid, pCTX-M360, classified as IncM2, isolated from a multidrug-resistant *K. pneumoniae* strain from a patient in an intensive care unit in China in 2003, comprised a pEL60-like backbone (the *rep* region of 100% identity to that of pCTX-M3) with the IS*Ecp1*-*bla*_CTX-M-3_ module inserted downstream of *pemK*, and with an insertion of Tn*2* downstream of *repA*, introducing the *bla*_TEM-1_ gene [47]. Remnants of a similar transposon, Tn*1*, also with *bla*_TEM-1_, can be found within the MRR of pCTX-M3 [12]. Therefore, pCTX-M360 could be viewed as an intermediate between the ancestor, which probably was pEL60-like (or pC15_002-like), and pCTX-M3, and the insertion of Tn*1* would then represent an early event in the evolutionary sequence leading to pCTX-M3. 

Yet another plasmid, pNDM-HK, with *bla*_NDM-1_ encoding the New Delhi metallo-β-lactamase, isolated from *E. coli* of clinical origin in Hong Kong in 2009, can be considered a member of the pCTX-M3 family, and is also classified in the IncM2 subgroup [48]. It does not contain the IS*Ecp1*-*bla*_CTX-M-3_ module, and within its MRR the module with the *bla*_NDM-1_ gene replaces the integron fragment located downstream of one of the IS*26* copies in pCTX-M3 [48]. Additionally, pNDM-OM, another plasmid with the *bla*_NDM-1_ gene, isolated from a clinical *K. pneumoniae* strain in the Sultanate of Oman in 2010, is identical to pNDM-HK, lacking only two insertion sequences present in the latter [49]; pNDM-OM was also classified as belonging to the IncM2-subgroup. 

All known plasmids with 100% identity of their replicon region to the corresponding sequence of pCTX-M3 (1658 bp, starting 74 bp upstream of *repC* through *repA*, comprising *repCBA* and the RNA I-encoding sequence), found in the GenBank database (as of 25 March 2021), should be considered members of the pCTX-M3 family. They were found in Enterobacteriaceae strains (Enterobacterales ord. nov.) associated with health care (*K. pneumoniae*, *E. coli*, *C. freundii, Serratia marcescens*, *Enterobacter cloacae*, *E. hormaechei, S. enterica* subsp. *enterica* serovar Senftenberg, *K. michiganensis*, *Enterobacter kobei,* and *Leclercia adecarboxylata*). Some were also isolated from farm animals (from *K. pneumonia*, *E. coli*, *E. cloacae*, and *S. enterica* subsp. *enterica* serovar Havana), and additionally, a few originated from bacteria from municipal wastewater (*K. pneumonia* and *E. cloacae*), and one from *Klebsiella aerogenes* (pEa1631) from wildlife. Altogether, these environments indicate a vast diversity of reservoirs of the pCTX-M3 family plasmids. The 39 members of this family are shown in Figure 3a to the exclusion of sequences coming from the same survey and differing only in single nucleotides. As described above in detail for pCTX-M3, in most family members the modules comprising genes conferring antibiotic resistance can be found in two locations: the MRR between the *rep* region and the *trb* genes, and the shorter one downstream of *pemK*. The MRRs of the pCTX-M3 family plasmids, shown in Figure 3b, comprise a striking variety of genes conferring resistance to:

β-lactam antibiotics—*bla*_TEM-1_, usually located in proximity of the *rep* region, *bla*_IMP-4_ located within the integron, and also *bla*_NDM-1_, *bla*_DHA-1_, *bla*_OXA-16_, *bla*_OXA-48_, and *bla*_IMP-34_;aminoglycosides—*armA (nbrB)* imported into the plasmid with IS*CR1* as a part of a composite transposon flanked by two IS*26* elements, *aac(6’)-Ib4* related to IS*26* and *aac(3)-IId* located within an integron, also *aac(6’)-Ib-cr*, *aadA1*, *aadA16*, *aadA2*, and *strAB;*macrolides—*mel*, *mph1*, and *mph2*, frequently IS*CR1* related;chloramphenicol—*catB3* located within the integron;quinolones—qnrB2 and aac(6’)-Ib-cr;sulphonamides and trimethoprim—*sul1* and *dfrA12*, located within an integron, and also *dfrA14* or *dfrA27*;quaternary ammonium compounds—*qacE* and *qacG2*;mercury—the *mer* operon;fosfomycin—*fosC2*;bleomycin—*ble*_MBL_.

The second location harbors *bla*_CTX-M-3_ (*bla*_CTX-M-15_ in one case) accompanied by IS*Ecp1*. Only in pEsST2350_SE_NDM, a complex structure with multiple transposase genes and antibiotic resistance genes was found in this region.

All plasmids but one, pIMP_HB623 from an *E. cloacae* clinical strain from China, range in size from 66 to 98.5 kb. pIMP_HB623 of 133.2 kb carries a novel combination of genes conferring resistance to imipenem and fosfomycin within its 73-kb MRR [50]. This is a complex structure apparently generated by multiple recombination events. In four of the pCTX-M3-family plasmids the *trb* region is inverted: plasmid 2 of *S. enterica* subsp. *enterica* serovar Senftenberg from the USA, plasmid 6 of *K. pneumoniae* isolated in a Chinese hospital, and two plasmids, RCS55_pI and RCS40_p, isolated from *E. coli* strains in France. The inverted sequence blocks are bordered by two IS*26*s. Apart from this inversion, RCS55_pI and RCS40_p are 99% identical to pCTX-M3 along the entire sequence. Interestingly, pB12AN_1 isolated from a clinical *K. pneumoniae* strain in China lacks the *trb* genes, indicating a loss of self-transfer abilities. In the vast majority of the pCTX-M3 family plasmids the IS*26* insertion sequences or their remnants are present at the borders of the sequence blocks that differentiate their MRR regions (Figure 3b). As in other plasmids, IS*26* elements are important players in the plasmid evolution of the pCTX-M3 family by participating in the acquisition of antibiotic resistance genes [51,52]. It is also worth mentioning that some of the plasmids shown in Figure 3 were accompanied in the host cell by plasmids of other incompatibility groups facilitating further recombination events.

## 5. Taxonomy of IncL/M Plasmids

Since the 1970s numerous other plasmids classified as IncM, IncL, or IncL/M have been isolated from clinically relevant bacteria from the family Enterobacteriaceae (Enterobacterales ord. nov.). Their characteristic feature is one or two MRR(s) grouping antibiotic resistance genes as well as mobile genetic elements or their remnants. The problem of their classification was well presented in papers introducing new classification strategies in 2015 [19] and 2021 [53]. Initially, as discussed by Carattoli et al. [19], two groups were distinguished within 17 IncL/M plasmids—IncL and IncM, shown to be compatible with each other. Then, two subgroups were further distinguished among the IncM plasmids, IncM1 and IncM2; they were incompatible but phylogenetically well separated. Recently, the phylogenetic relations of 148 plasmids from the former IncL/M group were analyzed by Blackwell et al. [53]. By comparing the nucleotide sequences of 70 plasmid backbones they distinguished five subgroups within the IncL group, L1–L5. Plasmids recognized previously by Carattoli et al. [19] as members of the IncM1 and IncM2 groups were divided into five subgroups of IncM1, a single subgroup of IncM2, and additionally, four other subgroups were separated, IncM3–IncM6, [53]. While pEL60 became the unique member of the IncM3 subgroup, pCTX-M3, pCTXM360, pNDM-HK, and pNDM-OM remained the members of the IncM2 subgroup.

## 6. Conclusions

pCTX-M3 is a representative of a large family of low copy-number plasmids that are segregationally stable. Its conjugation system enables transfer of plasmids to a broad range of recipients of Alpha-, Beta-, and Gammaproteobacteria, and even to some Firmicutes, while it replicates only in Enterobacterales (ord. nov.). The reservoir of the pCTX-M3 family plasmids, which confer resistance to a wide variety of antibiotics, was found worldwide in diverse environments. As a part of a bacterial horizontal gene pool, such plasmids are involved in the exchange and spread of antibiotic resistance genes and finally, when disseminated in relevant bacteria, they pose serious problems for the health of humans and animals, compromising the efficiency of therapies. Therefore, it is an urgent need to understand the regulation of their conjugation system, aiming to develop agents which could counteract or control the conjugative transfer of plasmids.

## 7. Patents

Dmowski, M.; Kern-Zdanowicz, I. The helper plasmid, bacterial strain and the broad host range system for plasmid mobilization and uses thereof. PL230884. 10 September 2012.

## Figures and Tables

**Figure 1 ijms-22-04606-f001:**
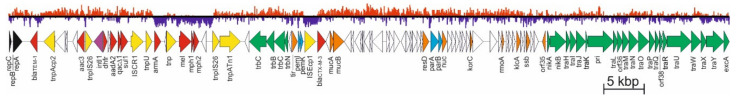
Schematic representation of pCTX-M3 sequence. Open reading frames (ORFs) are represented by arrows indicating their orientation. Genes encoding the replicon are in black, post-segregational killing and partition systems in blue, and conjugative transfer in green. Drug resistance genes are in red, transposase genes in yellow, other genes with homologues in databases in orange, and those of known functions are subscribed. A G+C plot is presented at the top; values higher than the average are in red and lower in purple.

**Figure 2 ijms-22-04606-f002:**
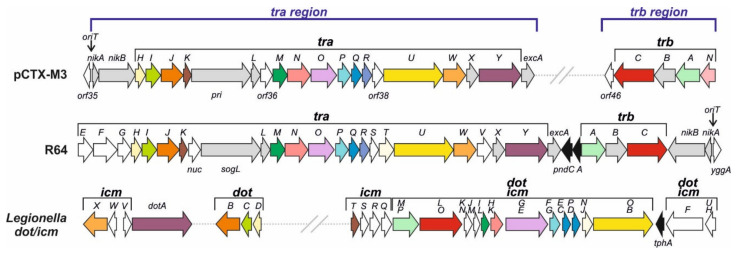
Gene organization of the transfer regions of pCTX-M3 and R64 and the *dot/icm* region of *Legionella pneumophila*. ORFs homologous in all systems are indicated by coloring. ORFs homologous in both plasmids are shown in grey, and those unique to one system only are shown in white. Gene group designation is indicated in brackets and individual genes are marked with appropriate letters. ORFs unrelated to the conjugation system are shown as black arrows.

**Figure 3 ijms-22-04606-f003:**
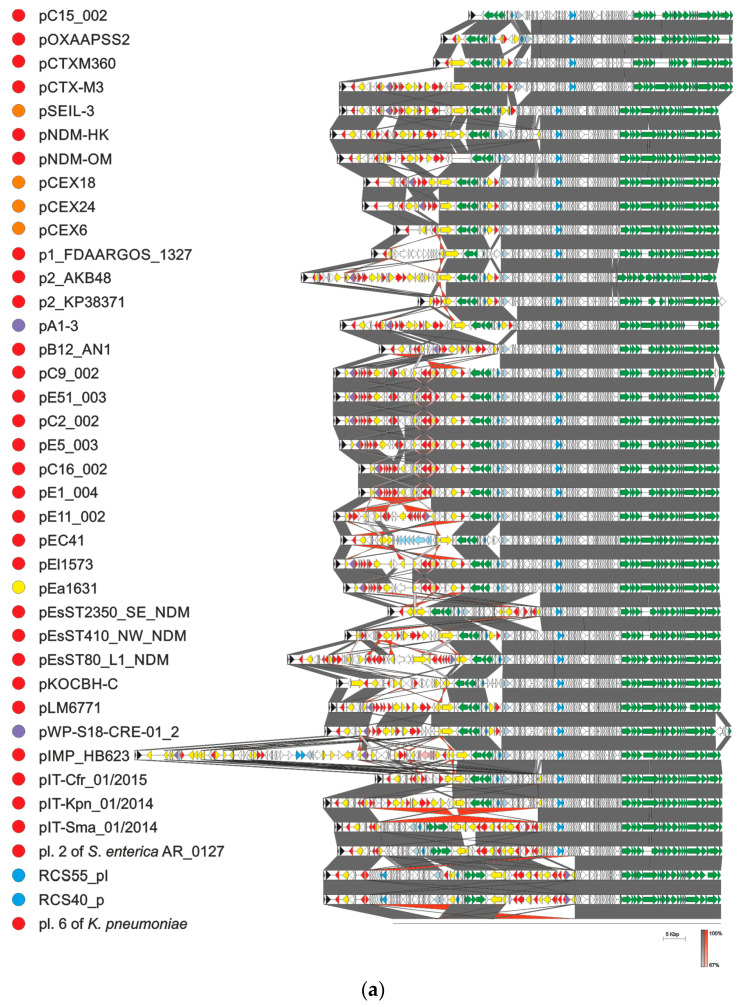
pCTX-M3 family plasmids. (**a**) Alignment of pCTX-M3 and related plasmids; (**b**) Alignment of diverse multiple resistance regions. The ORFs are represented as arrows, *rep* genes are in black, drug resistance genes in red, mercury-resistance operon in pink, ferric citrate transport genes in pale blue, conjugative transfer genes in green, transposase genes in yellow (IS*26* transposase genes are outlined in green in panel b), integron integrase-encoding genes in violet, *parAB* and *pemIK* in blue, *mucA* and *mucB* in grey, and others in white. Plasmid nucleotide sequences were taken from GenBank at following accession numbers: pC15_002—CP042490, pOXAPSS2—KU159086, pCTXM360—EU938349, pCTX-M3—AF550415, pSEIL-3—MN380440, pNDM-HK—HQ451074, pNDM-OM—JX988621, pCEX18—LC556221, pCEX24—LC556214, pCEX6—LC556219, p1_FDAARGOS_1327—CP069833, p2_AKB48—CP044337, p2_KP38371—CP014298, pA1-3—LC508263, pB12AN_1—CP026156, pC9_002—CP042532, pE51_003—CP042537, pC2_002—CP042522, pE5_003—CP042574, pC16_002—CP042580, pE1_004—CP042509, pE11_002—CP042526, pEC41—MW548582, pEl1573—JX101693, pEa1631—MG516907, pEsST2350_SE_NDM—CP031322, pEsST410_NW_NDM—CP031235, pEsST80_L1_NDM—CP031216, pKOCBH-C—CP035217, pLM6771—KX009507, pWP5-S18-CRE-01_2—AP022128, pIMP_HB623—KM877517, pIT-Cfr_01/2015—MH722216, pIT-Kpn_01/2014—MH722217, pIT-Sma_01/2014—MH722219, pl. 2 of *S. enterica* subsp. *enterica serovar Senftenberg* AR_0127—CP032193, RCS55_pl—LT985387, RCS40_pl—LT985241, and plasmid 6 *K. pneumoniae*—LT968692. All sequences were rotated or inverted and rotated to begin with *repC.* Plasmid schemes are drawn to scale. Original annotations are used. Color dots in front of the plasmid name denote the source of bacteria from which the plasmid was isolated: red—humans, orange—farm animals, yellow—wildlife, blue—humans or animals, and purple—wastewater. Sequence homology is color-coded according to scale below the alignment.

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
