# Peer review of "pCTX-M3—Structure, Function, and Evolution of a Multi-Resistance Conjugative Plasmid of a Broad Recipient Range"

_ijms, 2021, doi:10.3390/ijms22094606_

Round 1

Reviewer 1 Report

This is a well-written and comprehensive description of the plasmid pCTX-M3, which is an important vector for the spread of antibiotic resistance. It is a review article whose value is in the integration of information coming from a variety of studies. I found a few minor weaknesses:

  1. The abstract should specify the aspects of the plasmid that are covered in this review (this is totally absent from the abstract in its current form) and (if word limit permits) go over the main ideas. Some of these ideas are presented in (again if word limit permits) this section could be expanded.
  2. In Figure 3, the variable regions are too small to distinguish the details. It would be good to have a panel B showing only the MMR, or at least the largest one (between the rep region and the trb genes)
  3.  In line 408, the sentence "low copy-number plasmids segregationlly stable" should say " low copy-number plasmids that are segregationlly stable

Author Response

Response to comments of Reviewer 1

This is a well-written and comprehensive description of the plasmid pCTX-M3, which is an important vector for the spread of antibiotic resistance. It is a review article whose value is in the integration of information coming from a variety of studies.

I am grateful Reviewer 1 for his/her kind encouraging review.

I found a few minor weaknesses:

1. The abstract should specify the aspects of the plasmid that are covered in this review (this is totally absent from the abstract in its current form) and (if word limit permits) go over the main ideas. Some of these ideas are presented in (again if word limit permits) this section could be expanded.

I thank for this criticism; now I changed the text of the abstract (l. 9-20), adding the information describing more precisely the manuscript content.

2. In Figure 3, the variable regions are too small to distinguish the details. It would be good to have a panel B showing only the MMR, or at least the largest one (between the rep region and the trb genes)

I added the Figure 3B showing the differing MRRs regions, additionally I marked in it the IS26 transposase genes.

3. In line 408, the sentence "low copy-number plasmids segregationlly stable" should say " low copy-number plasmids that are segregationlly stable

The statement in line 408, now l 424, is corrected

Reviewer 2 Report

This review by Izabela Kern-Zdanowicz focus on the biology of pCTX-M3, an important multi-resistance plasmid. The manuscript is well-written, well organized, and presents adequate references. It was a pleasure reading this work, and get to know more about this plasmid. Congratulations to the author! I only have two minor suggestions:

  • Lines 222-223. Is this analysis performed by the author, or was performed previously? Anyway, some more details on the type of bioinformatics is important here.
  • Line 408. By 'low-copy number', does the author refer to a copy number similar to the chromosome, or smaller than that (i. e., only a fraction of the cells carry the plasmid)? Please clarify.

Author Response

Response to comments of Reviewer 2

This review by Izabela Kern-Zdanowicz focus on the biology of pCTX-M3, an important multi-resistance plasmid. The manuscript is well-written, well organized, and presents adequate references. It was a pleasure reading this work, and get to know more about this plasmid. Congratulations to the author!

I thank Reviewer 2 for her/his kind and encouraging comments.

I only have two minor suggestions:

  • Lines 222-223. Is this analysis performed by the author, or was performed previously? Anyway, some more details on the type of bioinformatics is important here.

This is the analysis presented in the paper Dmowski, Gołebiewski, Kern-Zdanowicz, 2018, J Bacteriol. The citation of this paper is now included to the text in lines 233-234 of the revised version of the manuscript;”Results of our bioinformatic analysis (for details see [40]) suggest that the T4SS of the pCTX-M3 conjugation system is composed of: (…)”

The bioinformatic analysis of theT4SS, encoded by pCTX-M3 is described in details in our J Bact, paper, therefore I decided to cite it in such “extended” way.

  • Line 408. By 'low-copy number', does the author refer to a copy number similar to the chromosome, or smaller than that (i. e., only a fraction of the cells carry the plasmid)? Please clarify.

I am grateful for this comment and I apologize that writing about the replicon I missed the information on the pCTX-M3 copy-number. Now, it is added to the manuscript in lines 121-123: “ In E. coli cells pCTX-M3 is present in approx. 0.5 — 0.7 copy per chromosome (measured in cells of early stationary culture), while a minireplicon, comprising the repCBA region, in 8 — 10 copies per chromosome [18]”.